# Dynamic Pullout Behavior of Multiple Steel Fibers in UHPC: Effects of Fiber Geometry, Inclination Angle, and Loading Rate

**DOI:** 10.3390/ma12203365

**Published:** 2019-10-15

**Authors:** Eun-Yoo Jang, Jung J. Kim, Doo-Yeol Yoo

**Affiliations:** 1Department of Architectural Engineering, Hanyang University, 222 Wangsimni-ro, Seongdong-gu, Seoul 04763, Korea; jangeu@hanyang.ac.kr; 2Department of Civil Engineering, Kyungnam University, 7 Kyungnamdaehak-ro, Masanhappo-gu, Changwon-si, Gyeongsangnam-do 51767, Korea; jungkim@kyungnam.ac.kr

**Keywords:** ultra-high-performance concrete, multiple steel fibers, fiber shape, inclination angle, loading rate, pullout behavior

## Abstract

This study examined the influences of fiber geometry, inclination angle, and loading rate on the pullout behavior of multiple steel fibers in ultra-high-performance concrete (UHPC). For this, two different steel fiber types, i.e., straight (S-) and hooked (H-), four different inclination angles (0°–60°), and four different loading rates (0.018 mm/s to 1200 mm/s) were considered. Test results indicated that the pullout performance of S-fibers in UHPC was improved by increasing the loading rate. The highest maximum pullout load of the S-fiber was obtained at the inclination angle of 30° or 45°. The maximum pullout loads of H-fibers also increased with increases in the loading rate, while their slip capacities rather decreased. No specific inclination angle was identified in the case of H-fibers that caused the highest maximum pullout load. The H-fibers yielded higher average bond strengths than S-fibers, but similar or even smaller pullout energies under the impact loads. The aligned S-fiber in UHPC was most sensitive to the loading rate compared to the inclined S-fiber and aligned H-fiber. The rate sensitivity became moderate with the fiber inclination angle. Consequently, the aligned S-fiber was recommended to achieve the best energy absorption capacity and interfacial bond strength at various impact loads.

## 1. Introduction

For several decades, concrete has been one of the most important construction materials because of its high compressive strength, durability, and low price. However, due to its inevitable drawbacks (i.e., low tensile strength relative to the compressive strength and high brittleness), steel reinforced concrete (RC) has been considered as a typical structural type. At the material level, numerous studies have been conducted to overcome the inevitable drawbacks of ordinary concrete by incorporating discontinuous steel, i.e., synthetic and natural fibers, which limit crack propagation and increase the crack width in accordance to the bridging effect. Particularly, the behavior of ordinary concrete including steel fibers has been previously studied [1,2,3,4,5]. In the mid-1990s, a special type of fiber-reinforced concrete, known as “reactive powder concrete,” was first introduced by Richard and Cheyrezy [6]. This special material is now referred to as ultra-high-performance fiber-reinforced concrete (UHPFRC), and it exhibits excellent compressive strength that is greater than 150 MPa and design tensile strength of 8 MPa in accordance with Association Française de Génie Civil (AFGC) recommendations [7]. These superb mechanical properties could be achieved by optimized granular sizes based on the packing density theory, low water-to-binder (W/B) ratio, special curing regime, and the high-volume contents of steel fibers. Correspondingly, it has attracted increased attention from researchers and engineers [8,9,10] for applications in protective structures under extreme loading conditions.

To understand the postcracking tensile performance of UHPFRC and the bridging capability of embedded steel fibers, the pullout behavior of these fibers in ultra-high-performance concrete (UHPC) matrix needed to be investigated. For this reason, several prior research studies [11,12,13,14,15,16,17] have performed static and dynamic pullout tests of various steel fibers embedded in UHPC. Additionally, efforts have been expended to identify the relationship between the pullout behavior of a single fiber and the tensile or flexural performance of composites [18,19,20]. Chan and Chu [15] have examined the effect of the amount of silica fume on the pullout resistance of aligned multiple straight steel fibers in UHPC. In their study [15], the amounts of silica fume ranged from 0% to 40%, and the enhancement of fiber-matrix interfacial properties was observed by incorporating the silica fume in the matrix. The optimum amount of silica fume was thus suggested to be between 20% and 30% based on the bond characteristics. In order to improve the pullout resistance of straight steel fibers, Wille and Naaman [14] have examined the effects of the sand ratio, silica fume, and glass powder on the pullout behavior of single aligned straight steel fiber in UHPC at different particle sizes and with and without nanosilica particles. They [14] reported some important findings that improved dispersion of fine particles based on enhanced particle packing and the use of nanosilica particles achieved a slip hardening response up to large slips, thereby causing very high-equivalent bond strengths beyond 20 MPa. Yoo et al. [16] evaluated the implications of using the shrinkage reducing admixture (SRA) on the pullout behavior of singly aligned, straight steel fibers in UHPC, and reported that its pullout performance in terms of the average and equivalent bond strengths and pullout work deteriorated with SRA increases owing to the reduced shrinkage of cement matrix. In addition, Kim and Yoo [17] have recently studied the effects of fiber shape and distance on the pullout behaviors of multiple aligned steel fibers in UHPC and reported the following useful information: (1) the order of average bond strength adheres to the order of twisted fiber > hooked fiber > straight fiber, (2) the bundled fiber exhibits the poorest bond performance in terms of bond strength and pullout work for all fiber types and the deterioration rate is most significant in the case of straight steel fibers, and (3) the average bond strengths of hooked and twisted steel fibers in UHPC are improved by decreasing the fiber distance, while the bond strengths of straight fibers are not influenced by it.

It is well known that since UHPFRC is a type of self-consolidating concrete, the arrangement of steel fibers can be changed according to the casting method and flow field conditions [21,22]. Therefore, to minutely analyze the fiber bridging capacity of UHPFRC, the pullout responses of aligned and inclined steel fibers in UHPC need to be understood. However, as explained above, most of the previous studies were carried out based on the aligned fiber specimens, and only a few studies [23,24] have conducted pullout tests using both aligned and inclined fiber specimens. Lee at al. [23] examined the effect of the inclination angle on the pullout behavior of multiple straight steel fibers in UHPC and reported that the highest maximum pullout load was obtained at the inclination angle of 30° or 45°, and that the slip capacity increased by increasing the angles up to 60°. In addition, Tai and Tawil [24] have recently studied the pullout behavior of singly aligned and inclined twisted, hooked, and straight steel fibers in UHPC under quasi-static and impact loads. They [24] found that the maximum pullout load and pullout energy of straight fiber increased by increasing the loading rate and inclination angle up to 45° in general, while less consistent trends were observed in the cases of hooked and twisted fibers. The order of rate sensitivity on the dynamic increase factor (DIF) adhered to the order of straight fiber > hooked fiber > twisted fiber.

Based on the test results of Kim and Yoo [17], the pullout behaviors of hooked and twisted steel fibers in UHPC were strongly influenced by the fiber distance that corresponded to the fiber volume fraction. Furthermore, Tai et al. [25] indicated that although their study verified the loading rate dependent pullout behavior of steel fibers in UHPC, other key parameters, such as fiber orientation, group effects, and fiber volume fraction, need to be analyzed to translate these pullout test results to the level of composites. However, unfortunately, to the best of our knowledge, there is no published study yet to comprehensively consider the effects of fiber shape, inclination angle, volume fraction, and loading rate on the pullout behavior of steel fibers embedded in UHPC.

Accordingly, this study investigates the pullout behaviors of multiple steel fibers embedded in UHPC at various inclination angles and loading rates within the range of 0.018 mm/s (quasi-static) to 1200 mm/s (impact). To achieve this, two different types of steel fibers (i.e., straight and hooked) and four different inclination angles of 0°, 30°, 45°, and 60° were considered, and the fiber distance was determined to correspond to the volume fraction of 2%. In order to rationally understand the pullout test results, failure modes, such as fiber pullout, rupture, and matrix spalling, were evaluated using image analyses, and the matrix spalling was quantitatively evaluated.

## 2. A Review on Dynamic Pullout Behavior of Steel Fibers in UHPC

Tai et al. [25] investigated the pullout behaviors of singly aligned, twisted, hooked, and straight steel fibers embedded in UHPC at various loading rates that ranged from 0.018 mm/s (static) to 1800 mm/s (impact). Based on the test results, they [25] reported that the twisted steel fiber exhibited the highest average bond strength and pullout energy, and the average bond strength and pullout energy of all types of steel fibers generally increased with increasing the loading rates. The highest sensitivity of the loading rate by the pullout resistance was observed in the straight steel fiber, followed by the twisted and hooked steel fibers. The pullout responses yielded progressively increasing the rate sensitivities at increasing pullout speeds and became significant at impact. Tai and El-Tawil [24] also evaluated dynamic pullout behaviors of single straight, hooked, and twisted steel fibers in UHPC with various inclination angles ranging from 0° to 45° and reported several important findings: (1) pullout resistance of straight fiber is maximized at the angles of 45° and 30° under the static and impact loads, respectively, (2) the straight fiber is most sensitive to the loading rate, following by the hooked and twisted fibers, (i.e., the dynamic increase factor (DIF) reached 1.93 at the impact for the aligned straight fiber and 2.32 for the 30°-inclined straight fiber, while the aligned hooked fiber had a peak DIF of 1.38 at the impact), (3) more matrix damage and fiber breakage occur in the deformed fiber cases, and (4) the energy dissipation capacity increases as the loading rate increases for all of the aligned steel fibers in UHPC in general. Kim et al. [26] evaluated the implication of loading rate on the pullout behavior of single hooked and twisted steel fibers embedded in normal- and high-strength cement mortar with compressive strengths ranging from 28 to 84 MPa. Their loading rates ranged were from 0.018 mm/s (static) to 18 mm/s (seismic). Based on test results, they reported that the hooked fiber shows no appreciable rate sensitivity when pulled out from the normal- and high-strength matrices, whereas the twisted fiber exhibits obvious rate sensitive pullout behaviors. The pullout energy of twisted fiber ranged from 1.90 to 5.15 times that of hooked fiber at all loading rates. Xu et al. [27] also examined the pullout behaviors of single straight, half-hooked, and low twisted steel fibers in UHPC under various loading rates ranging from 0.025 mm/s (static) to 25 mm/s (seismic) and reported several findings, as follows: (1) the half-hooked steel fiber provides the highest loading rate sensitivity on the maximum pullout load (DIF of 1.28 at the rate of 25 mm/s), followed by the straight (DIF of 1.19) and twisted (DIF of 1.14) fibers, (2) the twisted fiber is the most rate sensitive on the pullout energy (DIF of 1.43 at the rate of 25 mm/s), followed by the half-hooked (DIF of 1.24) and straight (DIF of 1.03) fibers, and (3) decrease in diameter increases the rate sensitivity of straight steel fiber in UHPC.

## 3. Research Significance

As summarized above, some researchers [24,25,26,27] have recently investigated the effect of loading rate on the pullout behavior of various steel fibers, singly embedded in UHPC. It is obvious that the pullout resistance, matrix spalling area, and rate sensitivity of steel fibers in UHPC are strongly affected by the number of fibers and their spacing, but there is no published study examining the dynamic pullout behavior of multiple steel fibers in UHPC yet. Thus, to take into account the actual implications of the fiber’s number and spacing on the dynamic pullout behavior, four steel fibers were embedded in UHPC matrix with a certain spacing corresponding to the volume fraction of 2% of commercially available UHPFRC and various inclination angles and pulled out under various loading rates ranging from 0.018 mm/s (static) to 1200 mm/s (impact).

## 4. Experimental Program

### 4.1. Specimen Manufacturing Process

There are many types of pullout tests depending on fiber type, number of fibers, loading rate, etc. In terms of the way the tensile force is applied to the specimen, there are two types of specimen tests, which include the single-sided and full-dog bone tests [23,26]. In order to reduce elongation, preliminary research minimized the free length in the single sided-test by using specially designed or reduced weight grips [23]. Single fiber tests are restricted to very small forces that are less than 50 N, which led to the calculation of the exact DIF and to the considerable increase of the uncertainty of the elicited results. Furthermore, single-sided tests can be used only for single fibers which do not consider group effects. In accordance to the reason stated above, this study will use double-sided pullout tests with four steel fibers based on the assumption of a 2% fiber incorporation rate. The distance between the fibers was calculated based on the assumptions of uniform distribution and perfect fiber alignment along the loading direction. The dimension of the specimen is illustrated in Figure 1.

Table 1 shows the chemical compositions and physical properties of cementitious materials that were used in this study. Type I Portland cement and silica fume are used to fabricate UHPFRC as cementitious materials. The detailed mix proportion that was applied is listed in Table 2. Silica sand with a maximum grain size below 0.5 mm was used as fine aggregate, and silica flour with a diameter of 2 μm and 98% of SiO_2_ was incorporated as a filler. In order to obtain excellent post-cracking tensile performance, the coarse aggregate was excluded from the mixture as usual [28]. As presented in Table 2, a very low water-to-binder (W/B) ratio value was used (0.2) to achieve a superb mechanical strength for UHPFRC. Owing to the low W/B ratio and increased number of fine constituents, the viscosity of the cementitious matrix was higher than that of the ordinary mixture, and led to low fluidity. In order to solve this problem and to achieve a self-consolidating property, a high-range water-reducing agent and a polycarboxylate-based superplasticizer (SP) were added that could increase the flowability.

First, all the compositions except liquid (mixing water and SP) were added into a Hobart-type mixer to fit its 5 L capacity and blended for 10 min at a low speed. Secondly, the mixed water with SP was poured into the mixer and was blended again for 10 min at a high speed. After the end-product assessments that included the compatibility of the mixture in terms of fluidity and viscosity, the fresh mortar was poured into the prepared molds. Since the specimen had a double-sided dog-bone shape, it needed to be poured two times in stages. For adopting various inclination angles of fiber properly, the steel fibers were fixed to the very thin polyvinyl chloride (PVC) sheet and foam board initially and its inclination angle was carefully checked by protractor. Cementitious mixture was poured in one side first and was kept for 24 h at room temperature to achieve adequate hardness. Subsequently, the foam board which was installed in the mold (opposite side) to fix the fibers was eliminated, and the mixture was poured to the other side and was hardened as described previously. After the mold was removed, the specimen was steam cured at a high temperature (90 °C) for 48 h to promote the development of strength and reduce the curing time compared to the case where curing occurred at room temperature.

Two different types of steel fibers (i.e., straight and with hooked-ends) were used to verify the effect of the steel fiber geometry on the pullout resistance. The geometrical and physical properties of the steel fibers used are summarized in Table 3. The diameters of hooked-end (H) and straight (S) steel fibers were 0.375 and 0.300 mm, respectively. Both the H- and S-fibers had identical lengths of 30 mm, that led to the aspect ratios (*l_f_*/*d_f_*) of 30/0.3 = 100 for S-fibers, and 30/0.375 = 80 for H-fibers.

### 4.2. Test Setup and Instruments

Five specimens were synthesized for each tested variable in order to obtain average values with minimum errors. In this study, two types of steel fibers were used, namely S- and H-fibers. Both fibers were fixed using the same embedded length to obtain a reliable outcome. The experimental variables also included the inclination angle (i.e., 0°, 30°, 45°, and 60°) and loading rate (i.e., quasi-static and impact loads). Consequently, 160 specimens were fabricated and tested in this study. The fiber volume fraction was fixed at 2% to consider the group effect between the four embedded steel fibers.

Two instruments were used in this study for static and dynamic testing. Figure 2 shows the schematic description of the static pullout behavior test setup. It was conducted using a universal testing machine (UTM) with a maximum load capacity of 3 kN. Uniaxial loads were monotonically applied to the specimen using the UTM. There was no need to consider the grip of the fiber because the full-dog bone shaped specimen was used, while a special grip that fitted well the dog bone specimens was used to grip them. The speed of the crosshead was controlled by software, and the loading rate of 0.018 mm/s was adopted for the quasi-static load, in compliance with prior published studies [23,26]. The pullout load was measured from a load cell affixed to the crosshead, and the slip was measured from the stroke displacement.

Dynamic fiber pullout tests were conducted using a specially fabricated impact test machine by domestic company, based on an air compressor and amplifier, as illustrated in Figure 3. The loading rate was controlled based on the pressure in the hydraulic air tank, and three different magnitudes of air pressures, i.e., 2, 5, and 8 kN, were applied. The dog-bone specimen was installed to the impact test machine using the grip, which was identical to that used for the quasi-static pullout tests. In order to obtain precise experimental results, a dynamic load cell with a responsiveness of 20 kHz was used that was higher than the responsiveness of common load cells. In addition, to achieve high sensitivity to measure the pullout load, the maximum load carrying capacity of the load cell was only 10 kN. The fiber slip from the cement matrix was measured using a potentiometer which was installed on the machine by the fixed magnet, as shown in Figure 3.

## 5. Experimental Results and Discussion

### 5.1. Determination of Loading Rate

The loading rate effect on the pullout behavior has been studied by previous researchers [12,29]. In their studies, the loading rate was calculated based on the 180-mm gauge length. For instance, a rate of 10^−1^ s^−1^ corresponds to a seismic loading rate. The loading rate was not constant, but it was varied with respect to time, even if constant air pressure was applied, as shown in Figure 4. Figure 4 shows a typical pullout load and displacement response for steel fibers in UHPC in the dynamic state according to time. As it can be observed, the slip of the specimen does not increase linearly. The slip of steel fibers in UHPC was gradually increased at the initial stage (up to the peak load), while a speedy increase of the slip was obtained after reaching the peak load. This might be caused by an easier pullout of fibers from the matrix. Owing to the nonlinear increase of the slip with time, it was very difficult to measure the loading rate at the peak load. Since it was difficult to find a publication which measured the exact loading rate in the dynamic states in the fiber pullout test, this task needs to be referred to a literature that measures strain. Wang et al. [30] has calculated a constant strain rate based on the ascending and descending points of the 80% peak strength to investigate the strain-rate-dependent compressive behavior of high-strength concrete. Similar to Wang’s study [30], the loading rate at the peak pullout load was calculated by taking an average between the 80% pre-peak point and post-peak point in this study. Despite the fact that data were acquired at a frequency of 20 kHz, the exact value equal to 80% of the peak load point could not be observed. Therefore, a linear interpolation method was used to obtain an accurate time and a displacement at the 80% point.

Figure 5 displays the effects of the fiber inclination angle and air pressure on the responses of displacement versus time. These results clearly show that the loading rate increased according to the inclination angle and applied air pressure. As the inclination angle increased (Figure 5a), the slip at the peak load was increased even at an identical air pressure, thereby leading to an increased loading rate. Since the slope of the slip and time curve—that indicates the loading rate (with units of ‘mm/s’)—increased with the slip (or displacement), a greater slip capacity resulted in larger loading rates. The slip at the peak is marked on Figure 5a, whereby larger inclination angles led to larger curve slopes. Increased air pressure also led to the expected increased rate of the temporal variation of the slip, as shown in Figure 5b. Therefore, the loading rate to the steel fibers in UHPC increased with increases in the applied air pressure.

### 5.2. Pullout Behavior of S-Fibers According to Inclination Angle and Loading Rate

Figure 6 shows the average pullout load versus slip curves of S-fibers embedded in UHPC according to the loading rate and inclination angle, and Table 4 summarizes all the parameters obtained from the fiber pullout tests. A quasi-static load with a loading rate of 0.018 mm/s was applied, and three different dynamic loads with air pressures set at 2, 5, and 8 kN were used. Four different fiber inclination angles set at 0°, 30°, 45°, and 60° were also used. The curves were drawn by averaging the test data obtained from at least three specimens for each variable, and different curve shapes were observed according to the loading rate and inclination angle.

According to previous research [23], the highest value of the maximum pullout load of the S-fibers in UHPC was obtained as the inclination angle that ranged between 30° and 45° under quasi-static states. Similar to previous studies [23], higher maximum pullout loads, *P*_max_, were obtained for the specimens, including S-fibers, with inclination angles of 30° and 45°, compared to the inclination angles of 0° and 60° in the static loading state. Among them, as summarized in Table 4, the highest maximum pullout load was obtained at of 45°, which equaled 375.7 N at the static loading rate of 0.018 mm/s. The aligned fiber with an inclination angle of 0° exhibited the minimum value of *P*_max_, regardless of the loading rate. For instance, the highest value of *P*_max_ of the specimen inclined at 45° was approximately 35% greater than that of the aligned specimen.

The highest maximum pullout loads were found to be 554.2, 612.2, and 646.6 N, respectively, at the applied air pressures of 2, 5, and 8 kN, respectively, when the inclination angle was 30° or 45°. Except for the case of the 5 kN pressure, the highest value of *P*_max_ was observed when the fiber inclination angle was 45°. Thus, the inclined S-fibers with the inclination angles of 30° and 45° more effectively resisted both the static and impact pullout loads as compared to the aligned and highly inclined S-fibers at 60°. When the S-fibers were inclined along the loading direction, a resistance force was obtained at the fiber exit, causing an additional resisting force [31,32]. Thus, the inclined S-fibers provided higher pullout resistance than the aligned fibers. However, once the fiber was significantly inclined in the direction of the pullout load, the matrix spalling was generated, and owing to a decrease of the embedment length by the spalling, a lower pullout fiber resistance can be obtained [23]. For these snubbing and matrix spalling effects, the highly inclined specimen at the inclination angle of 60° exhibited smaller *P*_max_ values as compared to the specimens with moderate inclination angles of 30° and 45°.

In order to quantitatively evaluate rate sensitivity on pullout resistance, the DIF was calculated using the maximum pullout loads of the static and dynamic states, as follows,
(1)$$\mathrm{DIF}=\frac{P_{d,\max}}{P_{s,\max}},$$
where *P_d_*_,max_ is the maximum dynamic pullout load, and *P_s_*_,max_ is the maximum static pullout load.

As given in Table 4, there was no obvious trend of the sensitivity of DIF to the loading rate according to the inclination angle. For instance, the highest values of DIF were found to be 1.50 and 1.72 for the 60° specimen at air pressures of 2 and 5 kN, while the highest DIF was 1.93 for the aligned specimen at 8 kN. In addition, the DIFs were only slightly different according to the inclination angle at the air pressures of 2 and 5 kN. Interestingly, the pullout load of aligned S-fibers also increased at increasing loading rates (Table 4). This result is consistent with the recent findings from Tai and El-Tawil [24], but inconsistent with the findings of Gokoz and Naaman [33]. Based on Gokoz’s study [33], the pullout behavior of straight steel fibers embedded in ordinary Portland cement mortar was almost insensitive to the loading rate. The clear increase of the *P*_max_ value of aligned S-fibers according to the loading rate was caused by an end-fiber deformation referred to as flattening. This deformation was elicited owing to the cutting process during fiber production [14]. The slight bending of S-fibers when they were installed in the molds could be also the reason for the increase of *P*_max_ value. Accordingly, although S-fibers were used, pure frictional bond stress was not activated alone, but instead, additional mechanical bond resistance was generated, thereby causing the sensitive behavior to the loading rate. Even for an air pressure of 8 kN, the highest value of DIF was obtained for aligned specimens because other specimens at different inclination angles exhibited a significant matrix spalling, thus limiting possible increases of the maximum pullout load.

The relationship between the slip capacity and the inclination angle is shown in Figure 7. The slip capacity denotes the slip at the peak pullout load, which is influenced by the loading rate. It was obvious that the slip capacity increased at increasing inclination angles, which was consistent with the findings of Lee et al. [23], but it was not influenced by the loading rate. A similar increasing slip capacity trend was observed according to the inclination angle for all static and impact loading conditions. No obvious slip capacity trend was also observed by Bindiganavile and Banthia [34] for the smooth steel fibers in accordance with the loading rate. The increased slip capacity was based on combined subbing and matrix spalling effects. As previously noted, the displacement of the test machine increased nonlinearly with the time of impact loading. The abrupt increase of the displacement was obtained at longer testing periods, as shown in Figure 5, so that the loading rates near the peak pullout load (80% of *P*_max_) for specimens with different inclination angles may be different, even if identical air pressure was applied. As was expected, the loading rate was greatly affected by the inclination angle at the identical pressure, as summarized in Table 4. Increased inclination angles resulted in higher loading rates, regardless of the magnitude of the air pressure, because the time needed to reach the slip capacity increased, as shown in Figure 5a.

To quantitatively evaluate the pullout behaviors of steel fibers in UHPC under both quasi-static and impact loads, the average and equivalent bond strengths, *τ_av_* and *τ_eq_*, and pullout work, *W_P_*, were calculated based on the following equations (Equations (2)–(4)).
(2)$$\tau_{av}=\frac{P_{max}}{\pi d_{f}L_{E}},$$
(3)$$\tau_{eq}=\frac{2W_{p}}{\pi d_{f}L_{E}^{2}},$$
(4)$$W_{P}=\int_{s=0}^{s=L_{E}}P\left(s\right)ds,$$
where *P*_max_ is the maximum pullout load, *d_f_* is the fiber diameter, *L_E_* is the initial embedment length, *s* is the slip, and *P*(*s*) is the applied pullout load with slip.

Figure 8 and Table 5 summarize the effects of inclination angles and loading rates on the pullout work, *W_p_*, the equivalent bond strengths, *τ_eq_*, and the average bond strength, *τ_av_* of S-fibers in UHPC. When the S-fibers were aligned in the direction of the pullout load, the values of *W_p_* and *τ_eq_* increased with loading rate increases that were caused by the increased maximum pullout load and slip capacity. This was consistent with the findings of Tai et al. [25] for singly embedded S-fibers in UHPC. For the quasi-static state, the highest pullout work and equivalent bond strength were obtained for the S-fibers at an inclination angle of 30°, followed by the inclination angles 45°, 60°, and the aligned specimens. Thus, it can be noted that owing to an increased maximum pullout load, slip capacity, or increases in both values, the inclined S-fibers were more effective in improving the energy absorption capacity under static pullout loads, compared to the aligned case, which is consistent with the single fiber pullout test results of Tai and El-Tawil [24]. Regardless of the fiber inclination angle, the pullout work and equivalent bond strength were all increased with increases in the loading rate, and the aligned fiber specimens were the most sensitive to the loading rate relative to other inclined fiber specimens. This was attributed to the significant matrix spalling observed in the inclined fiber specimens. Matrix spalling was generated by a stress concentration at the exit, and led to a sudden reduction of the embedment length. As given in Table 4 and Table 5, quite similar DIFs on the average bond strength were observed for all S-fibers, regardless of the inclination angle, but the larger inclination angle led to the faster loading rate. This indicates that the rate sensitivity on the average bond strength of S-fibers is influenced by the inclination angle. In addition, their pullout work and equivalent bond strength in Figure 8 were also noticeably affected by the fiber angle, whereby the aligned S-fibers were more sensitive to the loading rate on the bond strength and pullout energy than the inclined ones. Therefore, it is inferred that better fiber orientation along the loading direction is more effective in enhancing the post-cracking tensile strength and energy absorption capacity of UHPFRC under impact loads. This explanation can be supported by previous studies [35] that reported that higher post-cracking strength and energy absorption capacity in UHPFRC beams are obtained under impact loads when the S-fibers are more aligned and parallel to the direction of tensile load.

### 5.3. Pullout Behavior of H-fiber According to Inclination Angle and Loading Rate

Figure 9 shows the average pullout responses of multiple H-fibers in UHPC at various loading rates, while Table 4 summarizes all the parameters from the experiments related to the maximum pullout load. Five specimens were used to obtain the average curve, and the data are summarized in Figure 9 and Table 4 for each variable. Figure 9 was drawn in the same manner as the corresponding plot for the S-fibers. In the quasi-static state, only aligned fiber specimens exhibited a fiber pullout failure mode from the matrix (Figure 9a and Figure 10b). When the H-fibers were inclined toward the direction of the pullout load (30°), they were fractured before reaching to the ultimate bond resistance at the static state, as shown in Figure 10c. This was owing to the increased resistance at the exit of the inclined fibers that ultimately caused a higher bond resistance. In addition, matrix failure at the side surface of the half dog-bone specimen was also observed for the highly inclined H-fibers simultaneously (45° and 60°), as shown in Figure 10d. This was attributed to the low tensile strength and increased brittleness of the matrix. Thus, the surrounding matrix was easily fractured owing to the lateral load that is induced by pulling out the highly inclined fibers. At larger inclination angles, the ends of the fibers were positioned near the side surface, which led to the matrix failure at the inclination angles of 45° and 60° in the H-fibers. Robins et al. [36] had investigated the pullout behavior of hooked steel fibers according to the proportion of concrete mix, and reported that H-fibers tended to break the matrix as the inclination angle increased, regardless of the stiffness of the matrix. Therefore, the higher the angle of the H-fibers, the more frequent was the matrix breakage compared to the complete pullout or fiber fracture under quasi-static loads. The highest maximum pullout load of 962.8 N was observed when the inclination angle was 30°, which was due to a snubbing effect caused by the additional resistance induced at the exit of the inclined fibers. This is similar to the results of S-fiber cases [23]. The second highest value was obtained for the aligned specimen, followed by the specimens at the inclination angle of 45° and 60°. For instance, the highest maximum pullout loads of specimens inclined at 30° were approximately 15%, 26%, and 53% greater than the specimens inclined at 0°, 45°, and 60°, respectively. The reason for the decreased pullout resistance elicited by increasing the inclination angle from 30° to 45° or 60° was owing to the premature matrix failure. The premature matrix breakage limited the full development of the bond resistance of H-fibers in UHPC at a loading rate of 0.018 mm/s. Based on the same argument, Yoo et al. [13] had reported that high-performance fiber-reinforced cement composite beams, including H-fibers, exhibited the worst flexural performance among the various beams with S-fibers, despite their higher average bond strengths. Based on the examination of localized crack surfaces, they [13] found out that the end-hooks of most H-fibers were not straightened, and the crack surface was very rough. This was due to the matrix failure phenomenon of the inclined H-fibers observed in this study.

Interestingly, the aligned H-fibers in UHPC were all pulled out from the matrix, while the inclined H-fibers were all fractured before reaching the ultimate bond resistance at impact loads, as shown in Figure 9b–d. The relevant photographs on the typical failure modes of dog-bone specimens under impact loads are shown in Figure 11. Specifically, the highly inclined fibers at 45° and 60° also exhibited fiber fracture failure mode rather than the matrix failure (Figure 11b). This was attributed to the fact that the loading rate (or strain rate) sensitivity of the cracking matrix and fiber pullout were different. Yoo et al. [37] have recently noted that the initial cracking strength of cement matrix is more sensitive to the strain rate compared to the post-cracking strength of UHPFRC, which was closely related to the fiber-bridging capacity. This observation indicates that the increases of the bond strengths of steel fibers embedded in UHPC were smaller than those associated with the matrix cracking strength. Therefore, matrix failure was prevented under impact loads owing to the higher enhancement of the matrix strength compared to the fiber bond strength, even though the H-fibers were inclined towards the direction of the pullout loads. The highest maximum pullout loads were observed to be 938.5, 1026.8, and 1054.7 N, under the impact loads of 2, 5, and 8 kN, respectively. In addition, the highest maximum pullout loads were found at the inclination angles of 0°, 45°, and 45°, when the applied loads were 2, 5, and 8 kN, respectively. This was inconsistent with the findings of quasi-static results (Figure 9a), whereby the highest value was observed at the inclination angle of 30°. This inconsistency is attributed to the different failure modes according to the loading rate and fiber inclination angle.

As listed in Table 4 and shown in Figure 12, the average bond strength of H-fibers in UHPC generally increased with increases in the loading rate. This was consistent with the findings of several previous studies [24,26,34,38]. The H-fiber with an inclination angle of 45° exhibited the highest sensitivity to the loading rate, while that at the angle of 30° was least sensitive to the loading rate. For instance, the highest value of DIF specimens of 45° at the loading rate of 828.87 mm/s was found to be 1.38. The aligned and highly inclined (60°) H-fibers yielded similar sensitivities.

The higher loading rate sensitivities of H-fibers with inclination angles of 45° and 60° were elicited by the prevention of matrix failure under impact loads. In addition, when the H-fibers were inclined toward the direction of the pullout load, most of them were ruptured or exhibited matrix failure (Figure 10c,d). Thus, the average bond strengths of H-fibers generally decreased with increases in the inclination angle because of such a premature failure.

The slip capacity of H-fibers increased at increasing inclination angles, regardless of the loading rate, as shown in Figure 13. In addition, the slip capacity generally decreased with increases in the loading rate. For instance, the slip capacity of aligned H-fibers in UHPC was found to be 1.01 mm at the loading rate of 476.62 mm/s, which was approximately 82%, 76%, and 43%, of those at the loading rates of 438.58, 247.51, and 0.018 mm/s, respectively. This is consistent with the findings of Bindiganavile and Banthia [34] who reported that there was a general reduction in the slip capacity with increases in the loading rate for the deformed steel and polymeric fibers. This stiffening in the fiber and matrix bond up to the peak load was beneficial in limiting the widening of microcracks in the strain-hardening zone of UHPFRC, thus causing a higher post-cracking stiffness. Based on this prior study [34], such a higher stiffness in the fiber-matrix bond can elicit benefits from the serviceability point-of-view since a large crack width is not tolerated.

As shown in Table 5, the pullout work of H-fibers in UHPC decreased with an increase in the inclination angle at the quasi-static state. This was attributed to the premature matrix failure or fiber rupture before the ultimate bond strength was reached, as shown in Figure 10. In the case of highly inclined fiber, fiber was not straightened like aligned fiber, but keeps an original shape. For the case of aligned H-fibers, the pullout work and equivalent bond strength were not influenced by the loading rate. Thus, similar values of *W_p_* and *τ_eq_* were found that equaled about 4000 N·mm and 16 MPa for the aligned H-fibers at both the quasi-static and impact loads, ranging from 0.018 mm/s to 476.62 mm/s. The reason for this finding was attributed to the fact that although the maximum pullout load increased with an increase in the loading rate, the slip capacity was rather reduced and matrix spalling was generated, thereby causing a steeper decrease of the pullout load versus slip, as shown in Figure 9. As illustrated in Figure 11c, the matrix spalling became more obvious under the impact load and by increasing the loading rate. The effect of an increased maximum pullout load on the pullout energy absorption capacity was therefore offset by the reduced slip capacity and the steeper decrease of the post-peak pullout load with severe matrix spalling. The pullout work of the inclined H-fibers at 30° angles were also insignificantly affected by the loading rate, except for the case of the 2 kN pressure. Conversely, the pullout work of highly inclined H-fibers (45° and 60°) obviously increased with increases in the loading rate. This was caused by the change owing to the failure mode type change from matrix failure to fiber rupture as shown in Figure 11. Since the matrix failure occurred prior to fiber rupture, higher pullout energies were obtained under the impact loads. The values of *W_p_* and *τ_eq_* attained in the case of the highly inclined H-fibers approximately increased as much as two times by the pullout impacts. Therefore, it is concluded that there is insignificant loading rate effect on the pullout work and on the equivalent bond strength of H-fibers when an identical failure mode occurs. However, if the failure mode is changed from matrix fracture to fiber rupture, these values become sensitive to the loading rate.

### 5.4. Comparisons between Pullout Parameters of S- and H-fibers in UHPC

#### 5.4.1. Discussion on Bond Strengths and Pullout Work

Table 5 summarized the pullout test results of S- and H-fibers embedded in UHPC at various inclination angles and loading rates. It is obvious that the H-fibers exhibit much better pullout resistance at all loading rates in terms of the higher average and equivalent bond strengths, slip capacity, and pullout work, as compared to those of the S-fibers when these are aligned. This is owing to the end-hooks that induce an additional bond resistance along with the frictional component and consistent with the findings of Tai et al. [25] and Yoo et al. [13]. Although the inclined H-fiber prematurely failed by rupture of matrix failure (Figure 10c,d) before reaching the ultimate bond resistance, the average bond strengths of all H-fiber cases were higher than those of the S-fibers in UHPC. However, the differences between the bond strengths of H- and S-fibers became smaller at higher inclination angles and loading rates. For instance, the average bond strengths of aligned and inclined H-fibers at 60° under the impacts with an air pressure of 8 kN were found to be 22.1 and 16.4 MPa, respectively. These bond strength values were approximately 68% and 9% higher than those corresponding to aligned and inclined S-fibers at an angle of 60°. Thus, we could expect that higher post-cracking tensile strength of UHPFRC can be achieved with H-fibers compared to S-fibers only in the case where the fibers are well dispersed in the matrix and aligned along the loading direction. For this reason, in Wille’s study [39], similar post-cracking tensile strengths of UHPFRCs with 2 vol.% of S- and H-fibers were observed under both static and dynamic loads ranging from 10^−4^ s^−1^ to 10^−1^ s^−1^. For instance, the post-cracking tensile strength of 18.1 MPa was obtained equally for UHPFRCs with 2 vol.% of S- and H-fibers at the strain rate of 10^−1^ s^−1^. According to a previous study performed by Yoo et al. [13], even smaller post-cracking flexural strengths were found for UHPFRC beams with 2 vol.% H-fibers (H30) compared to 2 vol.% S-fibers (S30), and non-straightened H-fibers with an uneven crack surface were observed after a complete beam failure at the quasi-static state. This means that most of the H-fibers prematurely failed before reaching their ultimate bond strength. The reason for obtaining more noticeable deterioration of the post-cracking strength in the beam [13] compared to the dog-bone sample under tension [39] is possibly attributed to the poorer fiber dispersion and orientation in the latter compared to the former cases. It has been known that a better fiber orientation of UHPFRC is generally obtained in a smaller specimen owing to a more drastic gradient of flow velocity [21]. Therefore, the beam with a larger size compared to the dog-bone specimen had a poorer fiber orientation, i.e., more fibers are inclined along the direction of the load, thereby leading to the premature failure of H-fibers with poor post-cracking strength. Accordingly, it should be noted that fiber orientation needs to be carefully checked when the H-fibers are used with the UHPC matrix.

As indicated in Table 5, the aligned H-fiber exhibited significantly higher pullout work as compared to the aligned S-fibers at the quasi-static state. This was attributed to the higher maximum pullout load and slip capacity. However, interestingly, similar equivalent bond strengths of aligned S- and H-fibers in UHPC were obtained at impact loads, because of the reduced slip capacity and matrix spalling that occurred only in the case of H-fibers. This is inconsistent with the findings of quasi-static test results. Furthermore, owing to the premature rupture of inclined H-fibers or matrix fracture, even higher pullout work was observed for the S-fiber specimens at impact loads compared to H-fiber specimens. Thus, it can be noted that the use of S-fibers is more effective in enhancing the energy absorption capacity of UHPC at different load impacts compared to those of the H-fibers.

#### 5.4.2. Relationship between DIF and Loading Rate

The relationship between the DIF and loading rate for the bond strength of steel fibers in UHPC has not been adequately examined by previous studies. Figure 14 shows the relationships between the DIF values on the bond strengths of S- and H-fibers in UHPC at various inclination angles and loading rates. Only the samples that yielded fiber pullout failure modes were used to draw the relationship. The loading rate ranged from 0.018 mm/s to 1200 mm/s in this study. The investigation on the DIF and loading rate relation for interfacial bond strength of steel fibers in UHPC is important, because a post-cracking tensile performance of UHPFRC is closely related to the bond strength. Park et al. [40] evaluated the impact resistance of UHPFRC and notified its strain rate sensitivity under tension. They [40] reported that the tensile strengths of UHPFRC increased abruptly under tensile impact loads, and this dramatic increase was from the strain rate sensitivity of the bond strength of steel fibers, correlated to the microcracking behaviors.

In order to predict the relationship between the DIF and loading rate (or strain rate), many researchers have conducted rate sensitivity tests and proposed empirical models [41,42]. These models are well summarized in Salloum’s study [42]. The most empirical models were proposed for compressive and tensile behaviors of ordinary concrete or UHPFRC. Despite our efforts, we could not find any previously published studies that introduced the DIF-loading rate model for pullout behavior of steel fibers in UHPC. Thus, in order to compare more effectively the rate sensitivity of S- and H-fibers embedded in UHPC according to the inclination angle, the test data were simulated based on a simple linear function, as shown in Figure 14. The loading rates ranged from 0.018 mm/s to 500 mm/s for aligned S-fibers, and from 0.018 mm/s to approximately 700 mm/s, 1000 mm/s, and 1200 mm/s, for fibers with inclination angles of 30°, 45°, and 60°, respectively. The loading rate sensitivity on the bond strength was noticeably influenced by the fiber inclination angle and shape. The aligned S-fiber exhibited the highest sensitivity to the loading rate of DIF. Additionally, the rate sensitivity became moderate with increases in the inclination angle. Thus, the highly inclined S-fiber at an inclination angle of 60° was least sensitive to the loading rate. This is inconsistent with the findings of a previous study [23] that investigated the pullout behavior of single steel fibers in UHPC according to the loading rate. They [23] noted that the rate sensitivity in DIF was not significantly influenced by the inclination angle of the fibers. These conflicting results might be caused by the different specimen setups (single vs. multiple fibers). Most of the previously conducted pullout tests, including the previous study [23], have been conducted using single steel fibers which could not reflect the interference of surrounding fibers in actual composites. However, multiple steel fibers were used and embedded in UHPC matrix in this study, unlike previous studies, which led to significant spalling phenomena. The spalling of one random fiber influenced the remaining lateral fibers, thereby causing the decrease of the pullout load and the amplification of the matrix spalling. This phenomenon intensified as the loading rate increased. The extensive matrix spalling of multiple inclined fibers limited the increase of the interfacial bond strength according to the loading rate, as shown in Figure 15, relative to the aligned fiber case, thereby causing their diminished rate sensitivity. In addition, owing to the increase of the slip capacity of S-fibers in UHPC in accordance to the inclination angle, different loading rates were applied to the samples even though an identical air pressure was applied, as shown in Figure 5. The larger slip capacity led to greater loading rates at identical air pressures, possibly also affecting the rate sensitivity of S-fibers in UHPC in terms of the inclination angle.

As compared to S-fiber specimens, the H-fibers in UHPC exhibited decreased rate sensitivity to DIF, as illustrated in Figure 14. The matrix spalling limited the full development of the bond strength of H-fibers, thereby resulting in their lower rate sensitivity. In conclusion, it can be observed that the aligned S-fibers possess an excellent energy absorption capacity, as justified above, and also they are able to withstand higher impact loads compared to the static load and compared to the inclined S-fibers.

In order to obtain quantitative results on the spalling effect, image analysis was conducted as shown in Figure 15, Figure 16 and Figure 17. After taking a picture of the cross section at the front of the fiber exit, a part where the spalling was generated was adjusted to shade, and its portion among the whole image was analyzed using the program J-image. These image analyses were conducted only for the specimens that exhibited fiber pullout failure mode and no fiber breakage and matrix failure. Figure 15 shows the spalling area of S-fiber specimen at impact load according to inclination angle. This result can support both of the test results observed in the present study and previous studies [25] reporting that the maximum pullout load decreases when the angle increases. As shown in Figure 16, the H-fibers exhibit larger spalling areas due to their higher mechanical anchorage than the S-fibers, and the spalling area of H-fiber specimens was sensitive to the loading rate: the higher loading rate (or air pressure) led to the larger spalling area. On the other hand, the spalling area of S-fiber specimen was relatively insensitive to the loading rate. For instance, the spalling area of H-fiber specimen at impact states increases about 4.0 times as compared with that under quasi-static state, while the spalling area of S-fiber specimens increased only about 2.3 times under impact states, as shown in Figure 17b. However, although it is assumed that only shear friction is generated at the interface between the S-fiber and matrix, the S-fiber specimen showed the matrix spalling at all loading rates, including both static and impact. This was mainly caused by the fact that the S-fiber used was slightly curved at the original state (Figure 17a) and had ends deformed by the cutting process, which resulted in additional mechanical anchorage activation and matrix spalling.

## 6. Discussion on the Comparative Dynamic Pullout Behaviors of Single and Multiple Steel Fibers in UHPC

In order examine the implications of the number and spacing of fibers on the dynamic pullout behavior and rate sensitivity, the previous test results reported by Tai and El-Tawil [24] were compared because similar mix proportions and steel fiber types were applied. For the case of S-fiber, the highest maximum pullout load was observed when the inclination angles were between 30° and 45° at both the static and impact loading conditions regardless of the number of fibers. Under the static conditions, the H-fiber also showed the highest maximum pullout load at the inclination angle of 30° for both the single and multiple fiber cases, while the effect of inclination angle on the maximum pullout load became insignificant at the impact conditions. Thus, it can be denoted that the effect of fiber inclination angle on the maximum pullout loads of S- and H-fibers seemed to be not influenced by the number of fibers.

As shown in Figure 14, even though similar mix proportions and steel fiber types were adopted, the rate sensitivity on the DIF of maximum pullout load was varied according to the number of fibers. At the seismic loading rate of 18 mm/s, the single fiber specimens provided much higher rate sensitivity than the multiple fiber specimens for both the S- and H-fibers. On the other hand, the single S- and H-fiber specimens provided similar or slightly less rate sensitivity at the impact loading rates. To more rationally explain the influence of number of fibers on the rate sensitivity including the seismic range, a further study on the pullout behavior of multiple fibers in UHPC in the seismic range is required to be done. In addition, both the single and multiple S-fibers embedded in UHPC provided much higher rate sensitivity on the DIF of matrix pullout load than the H-fiber cases. Accordingly, it is denoted that the loading rate sensitivity on the pullout resistance is influenced by the number of fibers, but the trend of rate sensitivity according to the fiber type (straight vs. deformed) is not affected by it.

## 7. Conclusions

This study investigated the implications of fiber shape, inclination angle, and loading rate on the pullout behavior of multiple steel fibers embedded in UHPC. Four different loading rates from quasi-static to impacts that ranged from 0.018 to about 1200 mm/s were used to evaluate the relationship between the DIF and loading rate according to the fiber shape and inclination angle. The pullout test results were rationally explained based on the failure mode and image analyses, which quantitatively evaluated the matrix spalling area. From the test results and discussions, the following conclusions are drawn:(1)The maximum pullout loads of S-fibers increased with increasing loading rate, but no clear trend was elicited on the rate sensitivity of the DIF of bond strength according to their inclination angle. The slip capacity of S-fibers increased at increasing inclination angles but was not influenced by the loading rate.(2)The pullout work and equivalent bond strength of S-fibers increased with increases in the loading rate. However, the aligned S-fiber was more sensitive to the loading rate than the inclined ones due to the more severe matrix spalling phenomenon in the latter cases.(3)The maximum pullout loads of H-fibers in UHPC generally increased with increasing the loading rate. The H-fibers at the inclination angle of 45° exhibited the highest sensitivity to the loading rate, while the H-fibers at the angle of 30° were the least sensitive. Furthermore, the slip capacity of H-fibers increased with an increasing inclination angle but decreased with the loading rate.(4)Given the identical failure mode, the pullout work of the H-fibers in UHPC was insensitive to the loading rate, but became sensitive to the loading rate as the failure mode was changed from matrix fracture to fiber rupture.(5)The H-fibers generally exhibited higher average bond strengths than the S-fibers at all loading rates and inclination angles. However, similar equivalent bond strengths of aligned S-fibers and even higher pullout work of inclined S-fibers were observed at the impact as respectively compared to the aligned and inclined H-fibers.(6)The DIF of aligned S-fibers in UHPC was most sensitive to the loading rate. The rate sensitivity became moderate at increasing inclination angle of S-fibers. The aligned S-fibers are recommended to be used to achieve excellent pullout energy and interfacial bond strength at impact loading.

Yoo et al. [43] recently noted that there is difference between the fiber pullout and tensile test results of UHPFRC composites, mainly caused by random orientation and insufficient dispersion of fibers. The insufficient matrix volume covering the fibers in the composites potentially leads to the premature pullout failure of mainly deformed steel fibers from the UHPC mixture. Thus, even though useful information was achieved from the static and impact pullout tests of evenly distributed and oriented multiple steel fibers in UHPC, a further study is required to extend these results to the composite level.

## Figures and Tables

**Figure 1 materials-12-03365-f001:**
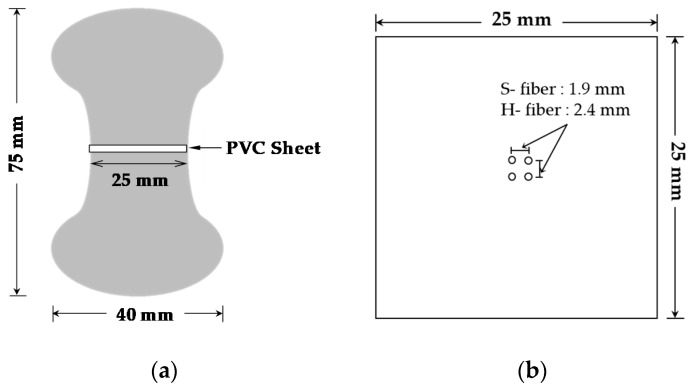
(**a**) Dimensions of the dog-bone specimen and (**b**) cross sectional details.

**Figure 2 materials-12-03365-f002:**
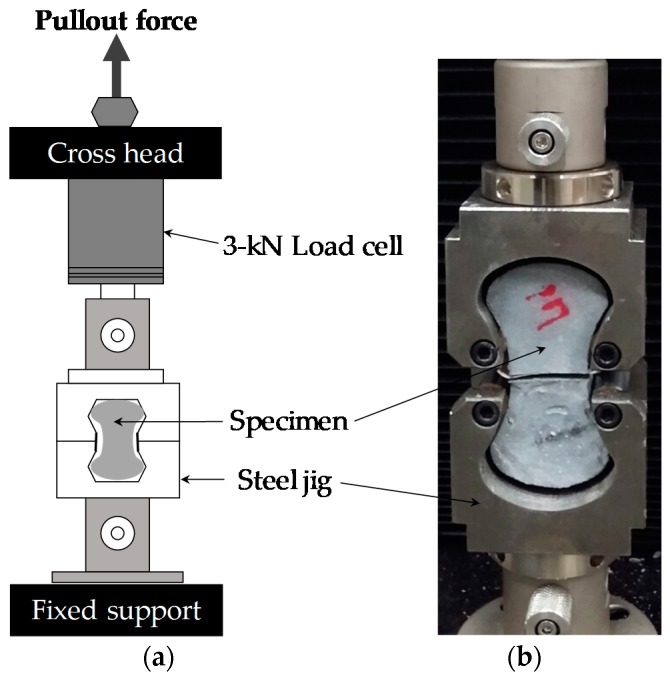
(**a**) Schematic of steel fiber static pullout behavior test setup, (**b**) grip details.

**Figure 3 materials-12-03365-f003:**
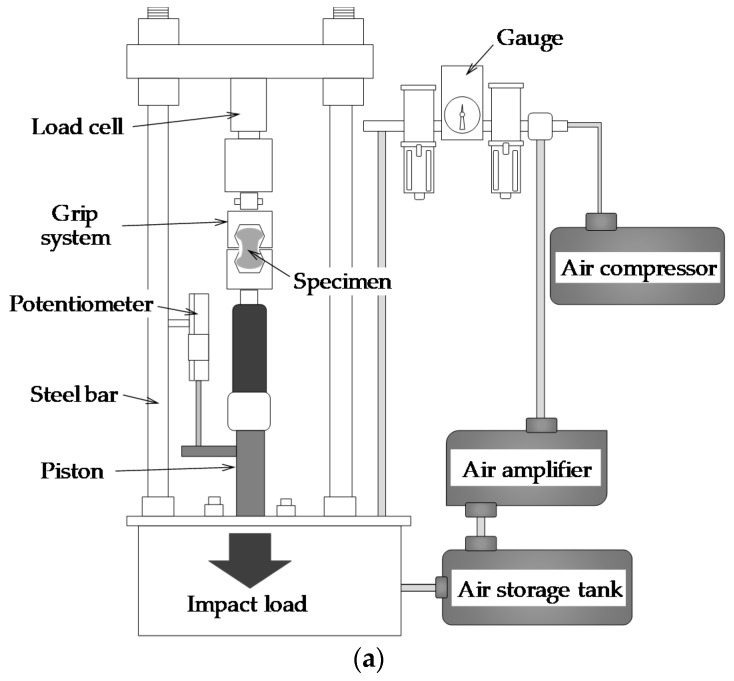
Fiber pullout impact test machine: (**a**) schematic description and (**b**) picture.

**Figure 4 materials-12-03365-f004:**
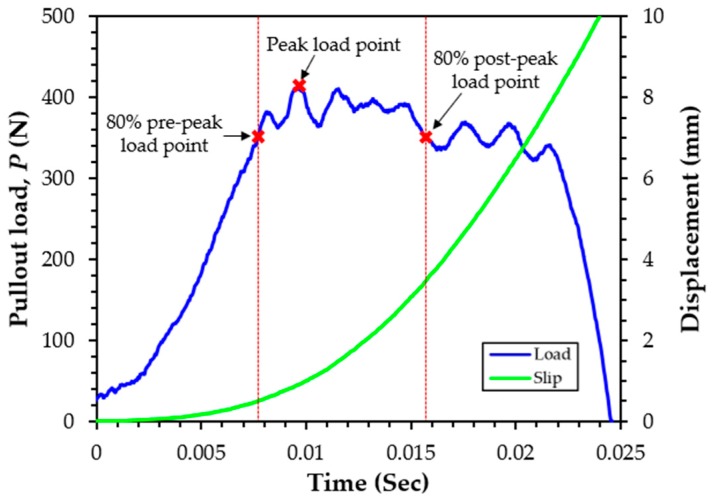
Measurement of loading rate from dynamic pullout test setup.

**Figure 5 materials-12-03365-f005:**
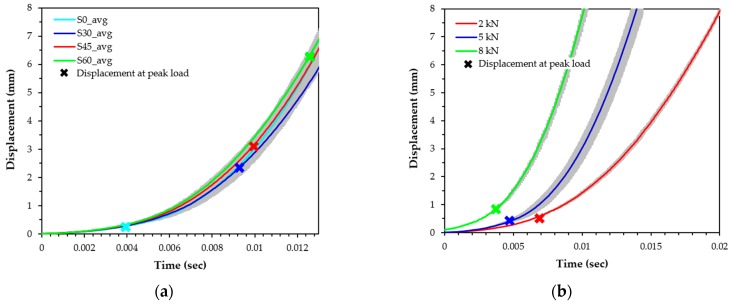
Time versus displacement curve: (**a**) displacement according to inclination angle at 5 kN, (**b**) displacement according to air pressure for aligned fiber.

**Figure 6 materials-12-03365-f006:**
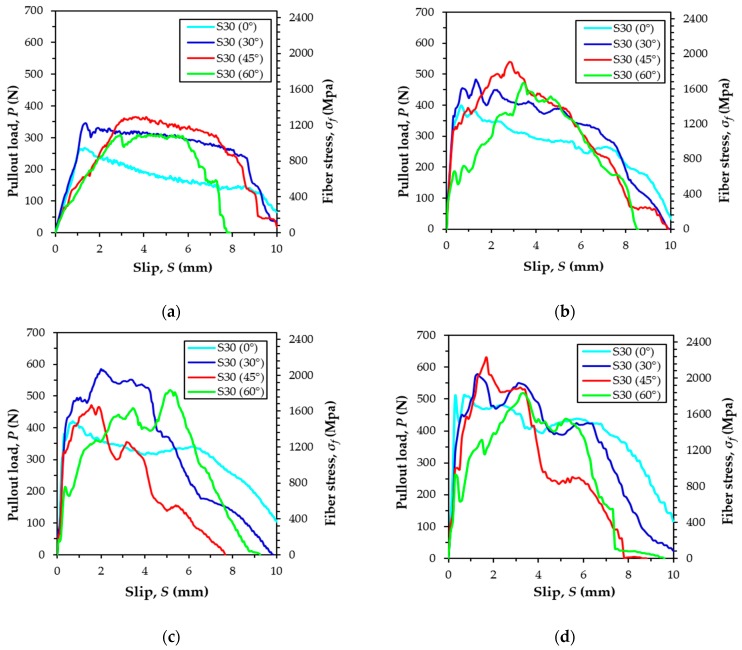
Average pullout load versus slip curves on straight (S)-fibers at various loading rates: (**a**) quasi-static, (**b**) air pressure of 2 kN, (**c**) air pressure of 5 kN, and (**d**) air pressure of 8 kN.

**Figure 7 materials-12-03365-f007:**
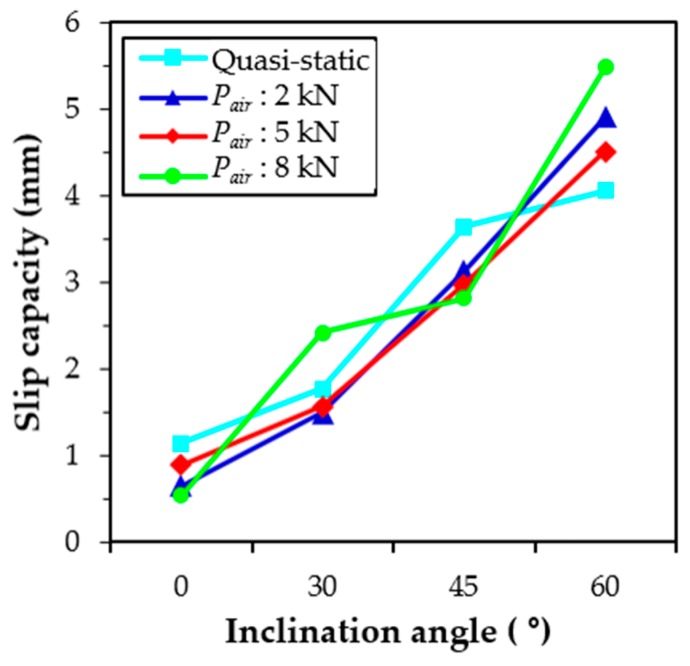
Effects of loading rate and inclination angle on slip capacity of S-fibers (Note: *P_air_* = air pressure).

**Figure 8 materials-12-03365-f008:**
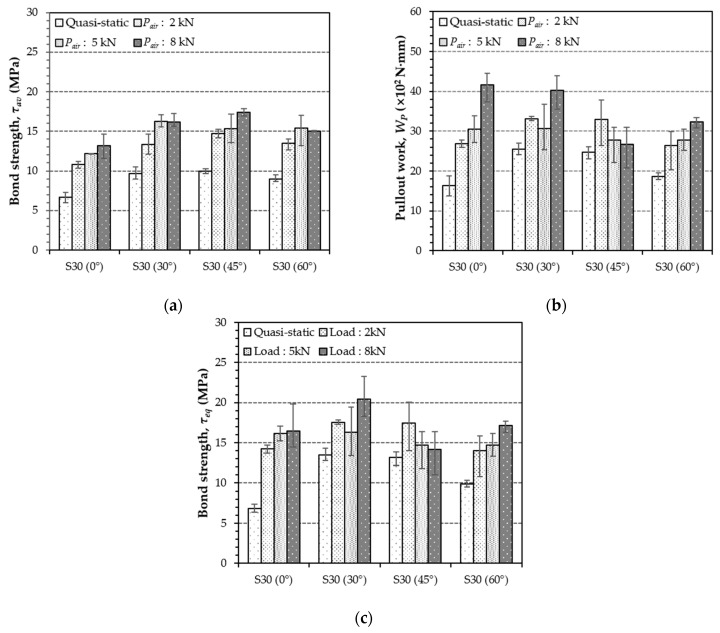
Summary of the effects of inclination angle and loading rate on pullout parameters of S-fiber in ultra-high-performance concrete (UHPC): (**a**) average bond strength, (**b**) pullout work, and (**c**) equivalent bond strength (Note: *P_air_* = air pressure).

**Figure 9 materials-12-03365-f009:**
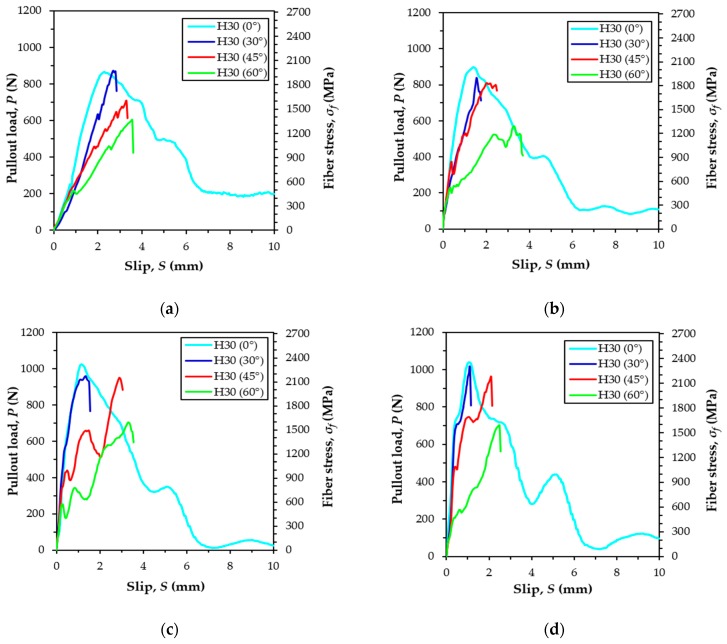
Average pullout load versus slip curves on H-fibers at various loading rates: (**a**) quasi-static, (**b**) air pressure of 2 kN, (**c**) air pressure of 5 kN, and (**d**) air pressure of 8 kN.

**Figure 10 materials-12-03365-f010:**
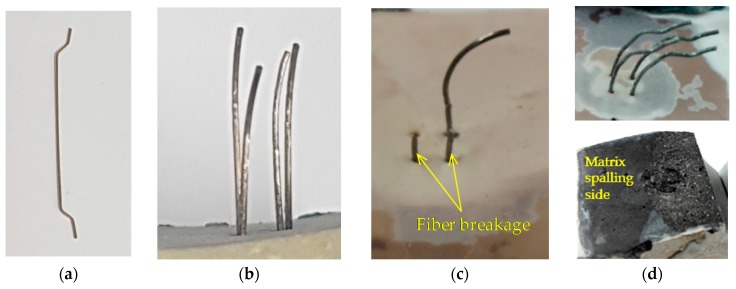
Pictures on various failure modes of H-fibers in UHPC: (**a**) original H-fiber, (**b**) aligned H-fiber, (**c**) H-fiber with an inclination angle of 30°, and (**d**) highly inclined H-fiber (i.e., 45° and 60°).

**Figure 11 materials-12-03365-f011:**
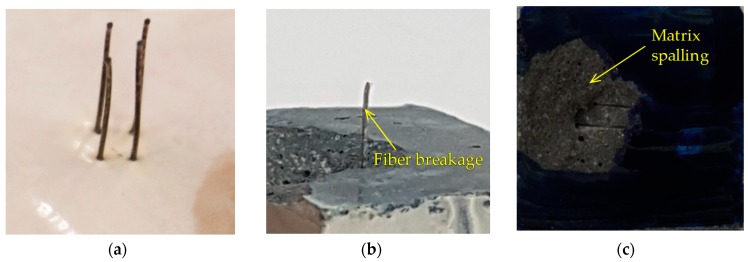
Failure modes of H-fibers according to inclination angle at impact loads: (**a**) aligned H-fiber, (**b**) inclined H-fiber, and (**c**) matrix spalling.

**Figure 12 materials-12-03365-f012:**
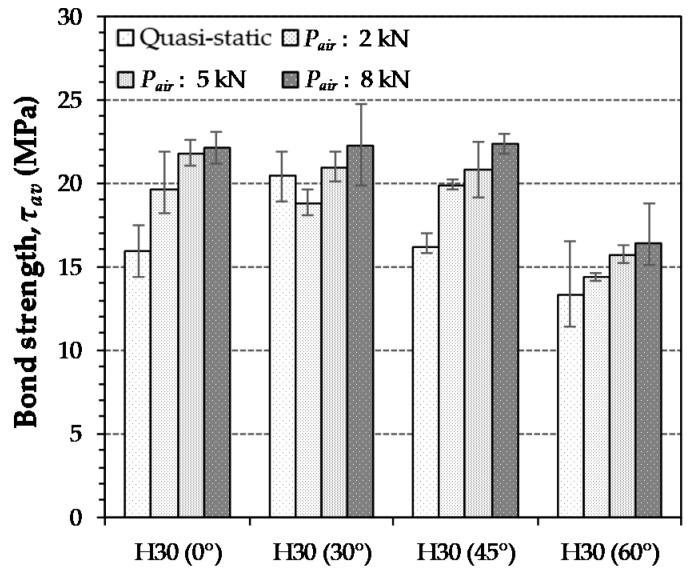
Effects of loading rate and inclination angle on average bond strength of H-fibers in UHPC (Note: *P_air_* = air pressure).

**Figure 13 materials-12-03365-f013:**
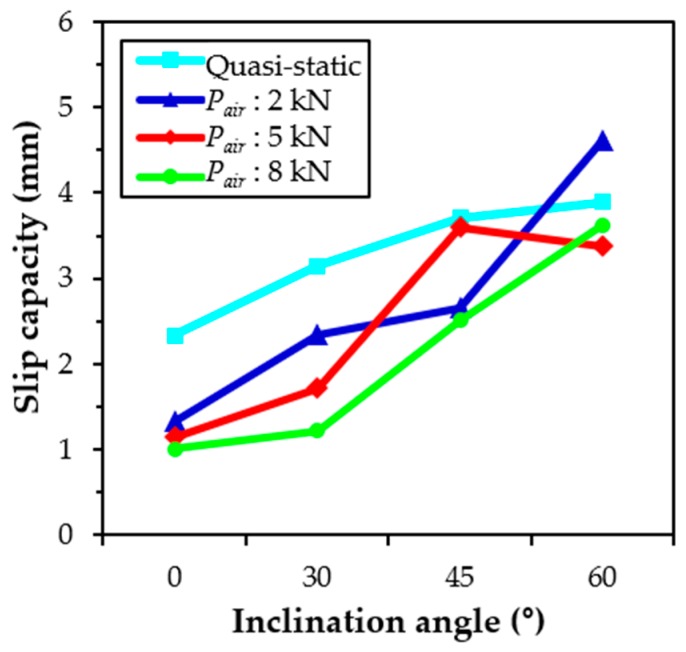
Effects of loading rate and inclination angle on slip capacity of H-fibers in UHPC (Note: *P_air_* = air pressure).

**Figure 14 materials-12-03365-f014:**
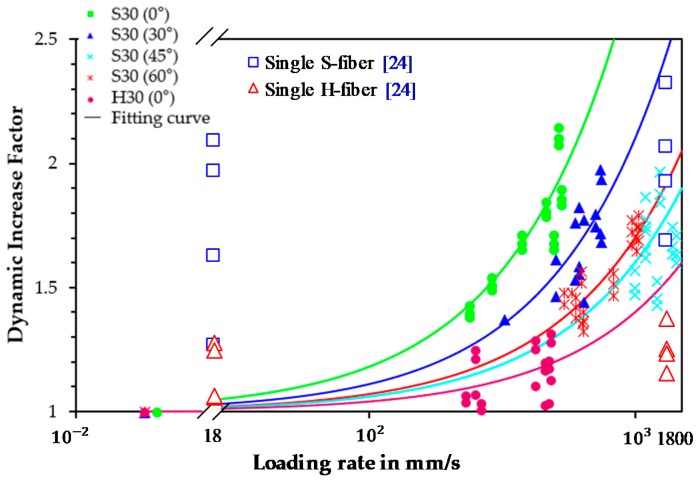
DIF versus loading rate relationship for S- and H-fibers in UHPC.

**Figure 15 materials-12-03365-f015:**
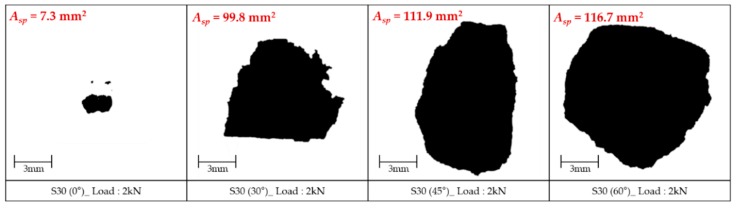
Spalling area of S-fiber sample according to inclination angle at air pressure of 2 kN.

**Figure 16 materials-12-03365-f016:**
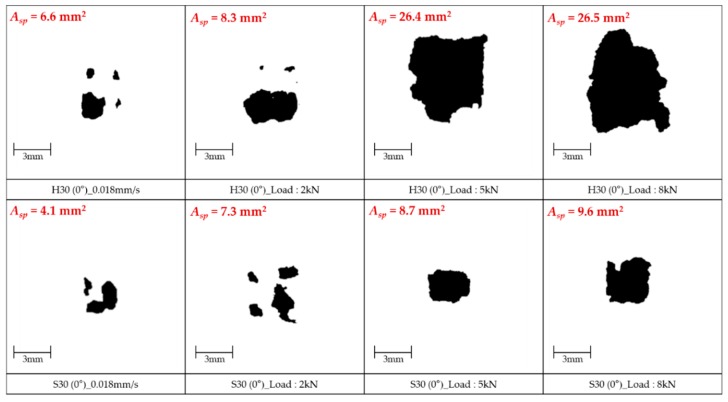
Spalling area of aligned S- and H-fiber specimens according to air pressure.

**Figure 17 materials-12-03365-f017:**
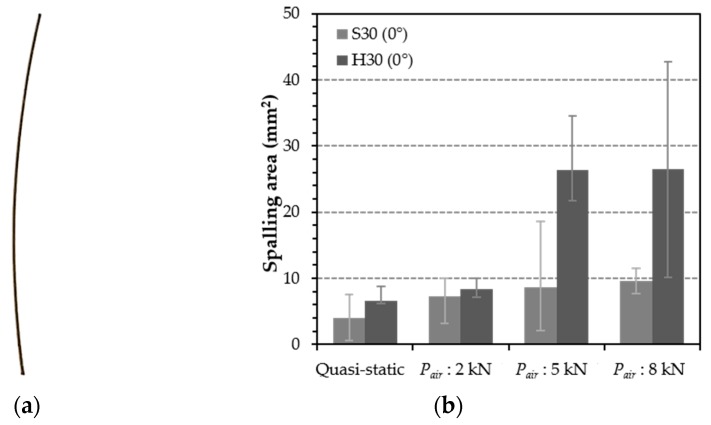
(**a**) Picture of a slightly curved S-fiber and (**b**) spalling areas of S- and H-fibers with various loading rates.

**Table 1 materials-12-03365-t001:** Compositions and properties of cementitious materials.

Composition % (Mass)	Type I Portland Cement	Silica Fume
CaO	61.33	0.38
Al_2_O_3_	6.40	0.25
SiO_2_	21.01	96.00
Fe_2_O_3_	3.12	0.12
MgO	3.02	0.10
SO_3_	2.30	-
Specific surface area (cm^2^/g)	3413	200,000
Density (g/cm^3^)	3.15	2.10
lg. loss (%)	1.40	1.50

**Table 2 materials-12-03365-t002:** Mixture proportions.

W/B	Unit Weight (kg/m^3^)					
	Water	Cement	Silica Fume	Silica Sand	Silica Flour	Superplasticizer
0.2	160.3	788.5	197.1	867.4	236.6	52.6

W/B = water-to-binder ratio. ^*^ Superplasticizer includes 30% solid and 70% water. ^†^ W/B is calculated by dividing total water content by total amount of binder.

**Table 3 materials-12-03365-t003:** Properties of steel fibers.

Type of Fiber	*d_f_* (mm)	*l_f_* (mm)	Aspect Ratio (*l_f_*/*d_f_*)	Density (g/cm^3^)	*f_t_* (MPa)	*E_f_* (GPa)
S30	0.300	30.0	100.0	7.9	2580	200
H30	0.375	30.0	80.0	7.9	2630	200

S30 = straight steel fiber with a length of 30 mm, H30 = hooked steel fiber with a length of 30 mm, *d_f_* = fiber diameter, *l_f_* = fiber length, *f_f_* = tensile strength, and *E_f_* = elastic modulus.

**Table 4 materials-12-03365-t004:** Summary of maximum pullout load and slip capacity of S- and hooked (H)-fibers at various loading rates.

Fiber Type	Inclination Angle	Quasi-Static				*P_air_*: 2 kN				*P_air_*: 5 kN				*P_air_*: 8 kN			
		*P*_max_ (N)	DIF	Slip*_u_* (mm)	Rate (mm/s)	*P*_max_ (N)	DIF	Slip*_u_* (mm)	Rate (mm/s)	*P*_max_ (N)	DIF	Slip*_u_* (mm)	Rate (mm/s)	*P*_max_ (N)	DIF	Slip*_u_* (mm)	Rate (mm/s)
S30	0°	278.5	1.00	1.14	0.018	406.7	1.46	0.65	262.5	458.9	1.65	0.54	268.7	537.7	1.93	0.89	499.0
	30°	363.0	1.00	1.77	0.018	503.9	1.39	1.49	410.6	612.2	1.69	2.41	648.4	596.3	1.64	1.56	662.6
	45°	375.7	1.00	3.63	0.018	554.2	1.48	3.13	568.7	576.5	1.53	2.82	708.9	646.6	1.72	2.98	999.5
	60°	338.0	1.00	4.06	0.018	507.7	1.50	4.90	736.6	581.9	1.72	5.48	1170.7	563.2	1.67	4.51	1293.9
H30	0°	837.3	1.00	2.33	0.018	922.8	1.10	1.33	247.5	1026.8	1.23	1.14	438.6	1041.5	1.24	1.01	476.6
	30°	962.8 ^†^	1.00	3.15	0.018	887.7 ^†^	0.92	2.34	371.3	917.0 ^†^	0.95	1.72	468.1	1050.3 ^†^	1.09	1.22	507.1
	45°	763.4 ^‡^	1.00	3.71	0.018	938.5 ^†^	1.23	2.66	416.0	981.5 ^†^	1.29	3.60	897.1	1054.7 ^†^	1.38	2.51	828.9
	60°	628.0 ^‡^	1.00	3.89	0.018	679.2 ^†^	1.08	4.62	624.1	741.5 ^†^	1.18	3.38	898.6	771.9 ^†^	1.23	3.63	1104.2

S30 = straight steel fiber with a length of 30 mm, H30 = hooked steel fiber with a length of 30 mm, *P*_max_ = maximum pullout load, *Slip_u_* = slip at the maximum load, and P_air_ = air pressure. ^†^ Fiber fracture; ^‡^ Matrix failure.

**Table 5 materials-12-03365-t005:** Summary of quasi-static and impact pullout test results.

Fiber Type	Incli. Angle	Parameters		Quasi-Static	*P_air_*: 2 kN	*P_air_*: 5 kN	*P_air_*: 8 kN
S30	0°	*τ_eq_*	MPa	6.84	14.22	13.92	16.49
			DIF	1	2.08	2.04	2.41
		*τ_av_*	MPa	6.63	10.79	11.33	12.33
			DIF	1	1.63	1.71	1.86
		*W_P_*	N·mm	1626.08	2680.72	3043.54	4154.05
			DIF	1	1.65	1.87	2.55
	30°	*τ_eq_*	MPa	13.49	17.56	16.30	21.32
			DIF	1	1.30	1.21	1.58
		*τ_av_*	MPa	9.63	13.37	16.24	16.36
			DIF	1	1.39	1.69	1.70
		*W_P_*	N·mm	2542.21	3310.51	3071.96	4018.84
			DIF	1	1.30	1.21	1.58
	45°	*τ_eq_*	MPa	13.14	17.48	14.72	14.16
			DIF	1	1.33	1.12	1.08
		*τ_av_*	MPa	9.96	14.70	15.29	17.39
			DIF	1	1.48	1.54	1.75
		*W_P_*	N·mm	2477.02	3294.46	2774.65	2669.41
			DIF	1	1.33	1.12	1.08
	60°	*τ_eq_*	MPa	9.85	13.98	14.69	17.12
			DIF	1	1.42	1.49	1.74
		*τ_av_*	MPa	8.97	13.47	15.44	14.99
			DIF	1	1.50	1.72	1.67
		*W_P_*	N·mm	1856.11	2635.05	2769.01	3227.95
			DIF	1	1.42	1.49	1.74
H30	0°	*τ_eq_*	MPa	15.53	14.62	16.24	16.14
			DIF	1	0.94	1.05	1.04
		*τ_av_*	MPa	15.96	19.59	21.79	22.10
			DIF	1	1.23	1.37	1.38
		*W_P_*	N·mm	4713.62	3444.82	3826.48	3801.96
			DIF	1	0.73	0.81	0.81
	30°	*τ_av_*	MPa	20.43 ^†^	18.84 ^†^	20.98 ^†^	22.29 ^†^
			DIF	1	0.92	1.03	1.09
		*W_P_**	N·mm	1468.17 ^†^	1153.24 ^†^	931.54 ^†^	837.65 ^†^
			DIF	1	0.79	0.63	0.57
	45°	*τ_av_*	MPa	16.21 ^‡^	19.92 ^†^	20.83 ^†^	22.38 ^†^
			DIF	1	1.23	1.29	1.38
		*W_P_**	N·mm	1376.87 ^‡^	1543.61 ^†^	1920.76 ^†^	1665.36 ^†^
			DIF	1	1.12	1.40	1.21
	60°	*τ_av_*	MPa	13.33 ^‡^	14.41 ^†^	15.74 ^†^	16.38 ^†^
			DIF	1	1.08	1.18	1.23
		*W_P_**	N·mm	1191.64 ^‡^	1686.90 ^†^	1448.57 ^†^	1571.32 ^†^
			DIF	1	1.42	1.22	1.32

S30 = straight steel fiber with a length of 30 mm, H30 = hooked steel fiber with a length of 30 mm, *τ_eq_* = equivalent bond strength, *τ_av_* = average bond strength, *W_P_* = pullout work, *W_P_** = pullout work up to the point where pullout load suddenly drops by fiber rupture or matrix failure. ^†^ Specimen fiber fractured; ^‡^ matrix breakage.

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
