# Peer review of "Dynamic Pullout Behavior of Multiple Steel Fibers in UHPC: Effects of Fiber Geometry, Inclination Angle, and Loading Rate"

_materials, 2019, doi:10.3390/ma12203365_

Round 1
Reviewer 1 Report
This paper focused on the influence of fiber shape, inclination angle, loading rate on the pullout behavior of steel fibers in UHPC. Many experiments were carried out and meaningful results were clearly presented. Therefore, I would like to suggest the acceptance of this paper after minor revision as follows:
1)'the implications of fiber geometry……‘ change into ' the influences of fiber geometry……'.
2) In the experimental part, how to control the inclination angle of fibers is not clearly given.
3) The conclusion part should be concise and mainly presents the new findings.
Author Response
This paper focused on the influence of fiber shape, inclination angle, loading rate on the pullout behavior of steel fibers in UHPC. Many experiments were carried out and meaningful results were clearly presented. Therefore, I would like to suggest the acceptance of this paper after minor revision as follows:
Answer: First of all, thank you so much for your valuable and useful comments on our paper. We have carefully considered all your comments. The revised manuscript is attached for your reconsideration. We would like to thank you for your comment which improved the quality of our paper.
1)'the implications of fiber geometry……‘ change into ' the influences of fiber geometry……'.
Answer: As you recommended, the word ‘implications’ is now changed to ‘influences’.
2) In the experimental part, how to control the inclination angle of fibers is not clearly given.
Answer: As you recommended, the detailed explanation on how we controlled the inclination angle is added in the experimental part, as follows.
“… For adopting various inclination angles of fiber properly, the steel fibers were fixed at the very thin PVC sheet and foam board initially and its inclination angle was carefully checked by protractor. Cementitious mixture was poured in one side first and was kept for 24 h at room temperature to achieve adequate hardness. Subsequently, the foam board which was installed in the mold (opposite side) to fix the fibers was eliminated, and the mixture …”
3) The conclusion part should be concise and mainly presents the new findings.
Answer: As you recommended, the conclusion part is now modified to be concise.
We sincerely appreciate again your useful comments on our paper again. We did our best to address all your comments. Please kindly and carefully take into account our answers above and reconsider your decision.
(Please see the attached file)
Reviewer 2 Report
The manuscript is properly prepared. The introduction is insightful, with references to other authors. Although it is a pity that the authors made little reference to European publications.
The research is well described. However, there is a lack of information about how the fibers were fastened in the specimens, especially since the fibers had different inclination angles. This information should be in the manuscript. It is also worth to include a suitable photo.
The used specimens are very small compared to the real fiber reinforced concrete elements. Do the authors consider the scale effect? It is worth referring to this issue in the manuscript.
The authors state that "this study investigated the implications of fiber shape, inclination angle, and loading rate on the pullout behavior of multiple steel fibers embedded in UHPC" what is significant and to some extent new. However, the research concerns the pullout behavior in which the fibers were positioned precisely and evenly spaced. This is of course necessary due to the transparency of the test performed, but in real fiber reinforced concrete elements the fibers are randomly distributed, inclined at different angles, the anchor lengths are different, the fibers form clusters (nests) or there are spaces without fibers. In connection with the above, it is worth considering to what extent the obtained results can be related to real fiber reinforced concrete structures. Authors should include information undertaking such discussion.
Author Response
The manuscript is properly prepared. The introduction is insightful, with references to other authors. Although it is a pity that the authors made little reference to European publications.
Answer: First of all, thank you so much for your valuable and useful comments on our paper. We have carefully considered all your comments. The revised manuscript is attached for your reconsideration. We would like to thank you for your comment which improved the quality of our paper.
The research is well described. However, there is a lack of information about how the fibers were fastened in the specimens, especially since the fibers had different inclination angles. This information should be in the manuscript. It is also worth to include a suitable photo.
Answer: As you recommended, the detailed explanation on how we fastened the fibers and controlled the inclination angle properly is added in the experimental part, as follows.
“… For adopting various inclination angles of fiber properly, the steel fibers were fixed at the very thin PVC sheet and foam board initially and its inclination angle was carefully checked by protractor. Cementitious mixture was poured in one side first and was kept for 24 h at room temperature to achieve adequate hardness. Subsequently, the foam board which was installed in the mold (opposite side) to fix the fibers was eliminated, and the mixture …”
Since we didn’t take photo for the installed fibers in the mold, a suitable photo is not able to be provided. So sorry.
The used specimens are very small compared to the real fiber reinforced concrete elements. Do the authors consider the scale effect? It is worth referring to this issue in the manuscript.
Answer: As you mentioned, there is some difference between the fiber pullout and tensile test results of UHPFRC composites1, which is mainly caused by random fiber orientation and insufficient dispersion. The insufficient matrix volume covering the fibers in the composites led to premature pullout failure of mainly deformed steel fibers from the UHPC mixture.
As you recommended, in order to give the above information to readers, the following sentences are additionally included in the text along with an additional reference1.
“Yoo et al. [43] recently noted that there is difference between the fiber pullout and tensile test results of UHPFRC composites, mainly caused by random orientation and insufficient dispersion of fibers. The insufficient matrix volume covering the fibers in the composites potentially leads to the premature pullout failure of mainly deformed steel fibers from the UHPC mixture. Thus, even though several useful information was achieved from the static and impact pullout tests of evenly distributed and oriented multiple steel fibers in UHPC, a further study is required to extend these results to the composite level.”
-----------------------------------------------------------------------------------------------------------------------------
[1] Yoo DY, Kim S, Kim JJ, Chun B. An experimental study on pullout and tensile behavior of ultra-high-performance concrete reinforced with various steel fibers. Constr. Build. Mater. 2019, 206, 46-61.
The authors state that "this study investigated the implications of fiber shape, inclination angle, and loading rate on the pullout behavior of multiple steel fibers embedded in UHPC" what is significant and to some extent new. However, the research concerns the pullout behavior in which the fibers were positioned precisely and evenly spaced. This is of course necessary due to the transparency of the test performed, but in real fiber reinforced concrete elements the fibers are randomly distributed, inclined at different angles, the anchor lengths are different, the fibers form clusters (nests) or there are spaces without fibers. In connection with the above, it is worth considering to what extent the obtained results can be related to real fiber reinforced concrete structures. Authors should include information undertaking such discussion.
Answer: Thank you so much for valuable comments and we surely agree with your opinion. Thus, in order to consider your suggestion, the following explanations on the differences between the fiber and tensile tests are added in the text, as follows.
“Yoo et al. [43] recently noted that there is difference between the fiber pullout and tensile test results of UHPFRC composites, mainly caused by random orientation and insufficient dispersion of fibers. The insufficient matrix volume covering the fibers in the composites potentially leads to the premature pullout failure of mainly deformed steel fibers from the UHPC mixture. Thus, even though several useful information was achieved from the static and impact pullout tests of evenly distributed and oriented multiple steel fibers in UHPC, a further study is required to extend these results to the composite level.”
We sincerely appreciate again your useful comments on our paper again. We did our best to address all your comments. Please kindly and carefully take into account our answers above and reconsider your decision.
(Please see the attached file)